# Design, Manufacturing Technology and In-Vitro Evaluation of Original, Polyurethane, Petal Valves for Application in Pulsating Ventricular Assist Devices

**DOI:** 10.3390/polym12122986

**Published:** 2020-12-15

**Authors:** Roman Major, Maciej Gawlikowski, Marek Sanak, Juergen M. Lackner, Artur Kapis

**Affiliations:** 1Institute of Metallurgy and Materials Science, Polish Academy of Sciences, Reymonta Str. 25, 30-059 Cracow, Poland; 2Department of Biosensors and Processing of Biomedical Signals, Faculty of Biomedical Engineering, Silesian University of Technology, Roosevelt Str. 40, 41-800 Zabrze, Poland; 3Artificial Heart laboratory, Foundation of Cardiac Surgery Development, Wolnosci Str. 345, 41-800 Zabrze, Poland; arti@frk.pl; 4Department of Medicine, Jagiellonian University Medical College, Skawinska Str. 8, 31-066 Cracow, Poland; nfsanak@cyf-kr.edu.pl; 5Joanneum Research Forschungsges.m.b.H., Institute of Surface Technologies and Photonics, Functional Surfaces, Leobner Str. 94, 8712 Niklasdorf, Austria; juergen.lackner@joanneum.at

**Keywords:** petal valve, ventricular assist device, blood–material interaction

## Abstract

Minimizing of the life-threatening thrombo-emboli formation in pulsatile heart assist devices by a new biomimetic heart valve design is presently one of the most important problems in medicine. As part of this work, an original valve structure was proposed intended for pneumatic, extracorporeal ventricular assist devices. The valve design allows for direct integration with other parts of the pulsating blood pump. Strengthening in the form of the titanium or steel frame has been introduced into the polyurethane lagging, which allows for maintaining material continuity and eliminating the risk of blood clotting. The prototype of the valve was made by the injection molding method assisted by numerical simulation of this process. The prototype was introduced into a modified pulsating, extracorporeal heart assist pump ReligaHeart EXT (developed for tilting disc valves) and examined in-vitro using the “artificial patient” model in order to determine hydrodynamic properties of the valve in the environment similar to physiological conditions. Fundamental blood tests, like hemolysis and thrombogenicity have been carried out. Very low backflow through the closed valve was observed despite their slight distortion due to pressure. On the basis of immunofluorescence tests, only slight activation of platelets was found on the inlet valve and slight increased risk of clotting of the outlet valve commissures as a result of poor valve leaflets assembling in the prototype device. No blood hemolysis was observed. Few of the clots formed only in places where the valve surfaces were not smooth enough.

## 1. Introduction

Mechanical circulatory supporting has become a routine life-saving therapy for patients with end-stage heart failure. Currently, there are two types of mechanical heart support in clinical practice: ventricular assist devices (VADs) and total artificial heart (TAH) [1,2]. VADs are a mechanical blood pump, used for heart support in the case of heart insufficiency. TAH are used to completely replace a biological heart with a mechanical device.

The first implantations of pulsating ventricular assist devices that mimicked the work of a biological heart took place in the 80s of the 20th century [3]. These devices were mainly used as a bridge to heart transplantation and the longest supporting times were up to 2 years. The development of medical knowledge and progress of technology resulted in the introduction to clinical practice at the beginning of the 21st century of rotary blood pumps, operating on the principle of an axial or centrifugal pump [4,5]. In contrast to pulsating pumps, these devices generate a non-physiological, continuous blood flow with extremely low pulsation. Despite this, the negative influence of long-term continuous flow on the human body has not been clearly proven [6,7]. Currently, rotary blood pumps are used both as a bridge to heart transplantation and as a destination therapy [8]. In very few cases the heart recovery may occur. Rotary blood pumps are much more convenient to use than pulsating pumps because they are small in size, are implanted inside the chest, and consume little electrical power. However, it has been unequivocally demonstrated that rotary blood pumps damage the von Willebrand protein, leading to life-threatening gastrointestinal bleeding. The design of these pumps contains gaps of several dozen micrometers, which is the cause of thrombotic complications, especially in patients with poorly selected anticoagulant and antiplatelet therapy.

The cost of a rotary blood pump is approximately 10 times that of a pulsating pump. This is important when a medical decision is required for a seriously ill patient. In such a case, first, a cheap pulsating pump is used, and after the patient’s clinical state improvement is achieved, a centrifugal pump is implanted in order to assure long term supporting. Another area of application of pulsating blood pumps is the treatment of heart failure in pediatric patients, especially up to several years of age, because centrifugal pumps are too large to be implanted in a child. Therefore, after 10 years of rotary blood pumps domination, engineers and physicians turns into the construction of upgraded and advanced pulsating blood pump, equipped with a modern kind of valves [9].

Currently, there are two manufacturers of pulsating heart support systems on the medical market (Berlin Heart Inc., Germany [10] and F.R.K. Intracordis, Poland [11]), including one manufacturer of pediatric pumps (Berlin Heart Inc., Germany) and two pulsating, total implantable artificial hearts (Syncardia, USA [12] and Carmat, France [13]). Proper operation of pulsating blood pumps requires the use of valves that direct the blood flow and prevent it from regurgitating (backflow). In pulsating heart prostheses valves made of artificial materials (e.g., titanium alloy and pyrolytic carbon [14]) are mainly used, with the exception of the Carmat artificial heart in which unique biological valves are used [13]. Except VADs and TAHs, since the 60s and 70s of the 20th century mechanical and biological valves have been used in cardiac surgery.

The correct selection of an artificial valve for use in a pulsating heart prosthesis is a complicated challenge and is crucial for the correct and safe operation of the device. In addition to the typical mechanical features such as reliability and durability, the valve must be constructed of highly biocompatible materials and its hydrodynamic effect on the blood stream must be minimal. The valve must be designed so as not to cause blood cavitation and high flow turbulence. The pressure drop across the open valve must be as low as possible (in the order of individual mmHg) and the reverse flow through the closed valve must be minimal (down to about 0.1 L/min).

Over the years, many mechanical valve designs have been developed for use in cardiac surgery. These valves have been reworked to enable their use in pulsed blood pumps. In general, mechanical valves can be divided into three basic types: caged ball (e.g., Starr-Edwards valve [15]), tilting disc (e.g., Biork-Shiley valve [16] or Hall [17] and bileaflet (e.g., ATS Open Pivot Mechanical Valve [18]). Caged ball valves consist of a silicone ball, a ring and three metal arches forming a cage made of titanium alloy or doped steel. This is the earliest construction of clinically applied artificial valve prosthesis. It is characterized by high durability, but their centrally closed construction results in a high value of pressure gradient and too high stress at its distal ends [19]. That is why presently these valves are out of use. Tilting disc valves are built with a single hinged disk made of pyrolytic carbon secured with a metal ring made of titanium alloy with a modified, athrombogenic layer [14]. In the open position, the disc is washed by blood, the angle of the disk opening is 60–80°, which gives two different sizes of outlets and matches mitral or aortic position. The construction of the bileaflet valve provides for two hemispherical leaflets, which opens to form one central and two peripheral holes to minimize blood flow disorders [20]. Currently, the structures of this type are the most frequently implanted mechanical valve prostheses. New arguments for the pro and against of bileaflet and tilting disc vales prostheses are still being brought forward, so the case remains open [21].

The second, large areas of application of mechanical valves are VADs and TAHs. It should be emphasized that in most cases, the mechanical valves used in them are typical valves used in cardiac surgery, which are only slightly modified to be installed in a blood pump. For pediatric VADs, commercial mechanical valves are too large in diameter, necessitating the design of new types of valves that are exclusive to this type of blood pumps. In this respect, polyurethane valves are used, made of a rigid, polymeric ring and flexible petals that deflect under the pressure of flowing blood [22]. The construction of this type of valve is complicated not only in terms of mechanical but also material. Mechanical damage of the valve consisting in rupture or detachment of the leaflet or insufficient stiffness of the leaflet will result in a partial or complete loss of the valve function as an element forcing unidirectional blood flow. The incorrect shape of the valve (both the ring and the leaflets) will adversely affect the structure of the blood flow, which will result in the formation of clots in the spaces poorly washed by blood. Similar thrombogenic effects and blood hemolysis will be observed in the case of insufficient biocompatibility of the materials used for the construction of the valve.

Besides low velocity of blood flow and poor washing regions, state-of-the-art heart supporting systems with mechanical valves generate platelet activating shear stress due to a narrow disc-ring gap. The listed factors lead to the most serious complication of mechanical circulatory support therapy, which are a thromboembolism and the resulting ischemic stroke.

The project goal is minimizing life-threatening thrombo-emboli formation, blood hemolysis and risk of cavitation in pulsatile heart assist devices intended for pediatric patients by applying a new biomimetic heart valve design based on metal–polymer composites and flexible leaflets.

## 2. Materials and Methods

### 2.1. Construction of a Single- and Bileaflet Flexible Inflow and Outflow Valve

The pulsating VAD has two valves: the inflow and outflow. These valves operate in different hydrodynamic conditions (pressure gradient, peak blood flow, and acceleration of blood movement), therefore they differ in design and size.

The inflow valve is of a single-leaf structure based on a set of two polyurethane elements—a rigid ring and an elastic petal, each of which is armed with a titanium frame (Figure 1). The valve leaflet has been designed as a deflecting flexible polyurethane element, which the shape ensures tight closure of the inflow channel lumen in the valve closed position and a small pressure drop while blood flows through an open valve. During the filling phase of the pump, the blood flowing into the blood chamber only slightly deflects the valve leaflet. Maintaining the continuity of the stream of blood flowing through the opened valve parallel to the wall of the blood chamber of the pumps promotes proper washing of the space around the leaflet on both sides, which prevents the formation of clots in these areas.

A fragment of the leaflet circumference is connected to the valve annulus by gluing with liquid polyurethane. The released edge of the leaflet, opposite to the glued part, is tilted so that the stream entering the blood pump is directed tangentially to the wall of the blood chamber of the pump. As a result of the deflection of the leaflet during the blood inflow phase to the pump, especially in the corner area, a vortex is formed, which washes the entire circumference of the membrane, the movement of which forces the blood to flow in the pump.

The only movable part of the valve subjected to long term work is the leaflet. To ensure its proper movement, it was necessary to strengthen the titanium or steel frame. It was designed as a composite leaflet with a thickness of 0.06 mm (Figure 2, on the left side). The titanium insert forms an elastic, open-shape frame supporting the leaflet, what determines its stiffness and degree of opening of the valve. At the same time, it protects the flexible leaflet in the closing phase against the pressure of the return wave, which could cause the leaflet to be pressed into the lumen of the inflow connector. The metal open-shape frame is embedded in the mass of the wall of the petal in a way that ensures adequate smoothness of the outer surface of the petal. The supporting element of the leaflet is the polyurethane ring—the valve housing (Figure 2, on the right side), the face of which has a shape compatible with the surface of the petal. In the closing phase, the valve leaflet is additionally mechanically supported by a seat made in the ring, which prevents the leaflet from bending under the pressure forces and additionally seals the leaflet-ring connection. The valve (both the ring and the leaflet) has been designed to be used in place of the inflow valve in the ReligaHeart EXT extracorporeal VAD [2].

The construction of the bileaflet inflow valve is a set consisting of three polyurethane elements: a rigid ring and two flexible leaflets, each of which is armed with an open-shape, titanium frame (Figure 3). The outflow valve leaflets are made of flexible polyurethane designed for long-term work (Figure 4a). Similarly to the inflow valve, the leaflets are armed with a flexible titanium frame in the form of a 0.06 mm open-shape frame. Same as in the inflow valve, it constitutes a flexible frame for the movement of hinged petals, and at the same time protects the flexible leaflets in the closing phase from a backing wave, which could cause their creasing and pressing into the lumen of the ring.

The valve ring has shaped sockets enabling fixing of the leaflets and shelves supporting the edges of the petals in the phase of full closure. The ring is equipped with a special titanium core (Figure 4b), which helps to stiffen the valve assembly and prevents it from deformation under applied pressure. The frame is completely embedded in the bulk of the material, except for the part of the mounting ring, which positions the valve assembly in the socket of its assembly in the pump. The openwork core design ensures translucency of the valve walls, enabling visual inspection of the inside of the valve.

At the bottom of the base of the leaflets, special technological retrograde flow channels were designed (Figure 5). The channel dimensions were 3 mm long and 1.5 mm wide. Their role is mainly in creating a side flow that flushes the bottom of the petals. The channels are equipped with miniature leaflets ensuring controlled blood flow in predetermined direction. This solution ensures a proper washing of the valve during its operation with blood in the most sensitive area—the bottoms of leaflets, protecting against the formation of thrombus due to the blood stagnation.

The technology of the valve leaflets manufacturing is based on the technology of high-pressure thermoplastic extrusion from biocompatible polyurethane BIONATE II. The valve leaflet is reinforced with a titanium micro mesh frame. The strengthening frame was additionally modified with a thin coating. For safety reasons and to increase the hemocompatibility of the whole leaflet, the frame was covered with a 95 nm a-C:H:N layer selected on the basis of several tests. This layer was characterized by a low coefficient of friction and protection of scaffolding delamination from TPU under cyclic load. Proper mechanical adhesion has been earlier confirmed in other laboratory tests and properties of scaffolding surfaces have been determined. The scaffolding was completely pressed in a polyurethane cover. Hybrid technology of a high-pressure thermoforming polymer: BIONATE II was adapted to make flexible TPU components.

### 2.2. Hemocompatibility Assessment

Expression of platelet activation was assessed by glycoproteins IIb/IIIa, using CD62P for P-selectin. The IIb/IIIa receptor is characteristic for the evaluation of the level of activation. It is an integrin on platelets. IIb/IIIa is the receptor for fibrinogen and the von Willebrand factor and helps in their activation. Complex IIb/IIIa is produced in natural conditions by an increase in calcium ions resulting from direct contact with a foreign factor, in this case a biomaterial. The most potent blood activator is adenosine-di-phosphate (ADP). It leads to a strong change in the conformation of platelet IIb/IIIa receptors and induces the binding of fibrinogen, a protein of blood plasma involved in the final phase of the blood coagulation process. The second analyzed platelet membrane receptor is selectin-P. P-selectin is a human protein encoded by the SELP gene. In inactivated plates, this protein is stored in granules of cells (in this case platelets) called Weibel-Palade. If the plates are activated, a membrane flip occurs. The receptors are released from granulation and exposure on the outer surface of the cell. After direct contact with the material, active plates and plate aggregates were analyzed. Platelet aggregates were analyzed by mixing 25 μL of blood with 0.4 mL of FLS and subsequent stabilization by adding 3.5 mL of 1% paraformaldehyde dissolved in PBS.

Activation of the coagulation system and nature of the immune response on the surface of the examined materials were analyzed using a laser confocal scanning microscope, Carl Zeiss Exciter 5. For this purpose, anti-CD62P and anti-CD45 monoclonal antibodies were used, respectively, to analyze the degree of activation of the coagulation system and immune response (defense tissues from a foreign body). The antibody finds a self-corresponding membrane receptor on the surface of cells. Monoclonal antibodies were conjugated with two different fluorochromes (anti-CD62P with fluorescein isothiocyanate (FITC), anti-CD45 with PE PE-Texas Red), which allowed distinguishing active platelets (green excitation) from active white blood cells (excitation red).

## 3. Results

This section may be divided by subheadings. It should provide a concise and precise description of the experimental results, their interpretation and the experimental conclusions that can be drawn.

### 3.1. Development of the Assembly Technology of the Flexible Inlet and Outlet Valve in the ReligaHeart EXT System

A modified structure of the ReligaHeart^®^ EXT extracorporeal heart assist pump was developed for the use of newly developed inflow and outflow valves (Figure 6). Original technical instrumentation intended for pump assembly by means of blood compatible materials was prepared. Tooling in the form of currently used injection molds was modified as well.

As part of the task, the technologies for the production of valve components and the technology of mounting the valves and the assembly of the valves in the ReligaHeart EXT extracorporeal heart support pump were developed. Both the inflow and outflow valve were glued into the sockets made in the pump parts in order to prevent blood leaking out of the pump housing.

### 3.2. Valve Elaboration Technology and Process of Valve Manufacturing

The construction of the injection molds for inflow and outflow valves were presented in Figure 7 and Figure 8, respectively. The experience gained with the manufacturing of regular ReligaHeart EXT blood pump and processing of commercial biomaterial BIONATE (manufactured by DSM Biomedical, Exton, PA, USA) was utilized.

Valve rings were made in two stages. In the first stage, metal stiffening cores were made with an openwork structure, having annular flanges fulfilling the function of positioning the valve in the booster pump. Valve cores were produced by machining technology—milling on a 5-axis computer numerical control (CNC) machining center. In the second stage, the polymer parts of valve rings were made by means of high-pressure injection technologies on injection molding machines.

Preliminary tests of injection molding were performed with the inserts made of stainless steel. They were overmolded with a polycarbonate urethane (PCU) with the trade name BIONATE II 55 D. The predrying of the polymer granules was carried out at +75 °C for 7 h. The insertion of the inserts and the removal of the overmolded components were done manually in the semiautomatic mode. The diameter of the hot runner injection was 1.8 mm for both components. The result of the technological process in the form of injected valve rings including stiffening cores was presented in Figure 9.

Leaflets for inflow and outflow valves were manufactured using high-pressure thermoforming technology from the polymer in the form of granules. Strengthening the petals in the form of an openwork frame was made of titanium foil. Laser cutting technology and finishing abrasive technology were used to fabricate the meshes to round the edges. Films were made with ironed mesh stiffening the leaflets, from which individual petals for the valves were cut out (Figure 10).

Valve assembly was carried using liquid solutions of a biocompatible polymer (BIONATE) based on the solvent (DMAc—dimethylacetamide). The solvent was the base material of both valve components. The assembly of valves and parts of ReligaHeart EXT VAD was carried out on automated computer-controlled assembly stations and with the use of a precision dispenser. Both valves and the blood chamber of the pump with mounted valves were presented in Figure 11.

### 3.3. Hydrodynamic Analysis in the Hydrodynamic Phantom Including Determination of the Risk of Turbulence and Wear

This part of research focused on understanding the hydrodynamic properties of composite valves as elements applicable in pulsed blood pumps.

The parameter that defines the usefulness of valves as blood flow directing elements is the retrograde flow (called backflow—BF). Measurements were carried out under reference, hydrodynamic conditions. The homogeneous nature of the flow was obtained by placing the tested valve in the center of the pipe with a length 20 times greater than its diameter. A liquid flow was generated in the direction closing the valve by means of a centrifugal pump BP80 (Medtronic, Minneapolis, MN, USA). The flow rate (BF (L/min), ±5%) was measured by transit-time, ultrasonic flowmeter (TS402, Transonic, Ithaka, NY, USA) and the pressure gradient across the valve (dp (mmHg), ±0.5%) was measured by means of piezoceramic gauges with analog voltage output (SML series, ADZ Nagano GmbH, Ottendorf-Okrilla, Germany).

The inflow valve with a titanium frame (abbreviation: ZNP-Ti) was tested in the range of dp = 20–300 mmHg. It was observed that the valve leaf bends under the action of pressure, which caused a permanent deformation of its geometry (Figure 12a). Nevertheless, the valve retained its backflow blocking properties as shown in Figure 12 (on the right side). Additionally, the resistance of the valve to damage under high pressure was checked. It was shown that dp = 760 mmHg did not damage the valve, but it significantly and irreversibly deformed the leaflet (Figure 12b). From dp = 400 mmHg, the leaflet deformation caused tightening of the valve ring lumen and a gradual decrease in the BF value (Figure 12, on the right side).

Next experiment compared the backflow vs. pressure characteristics of the valve in 3 cycles of operation: with the leaflet with undistorted geometry (cycle 1), the leaflet deformed by pressure in cycle 1 (cycle 2) and with the leaflet restored to its original geometry by manual straightening (cycle 3). Results were presented in Figure 13. The flow characteristics in cycles 2 and 3 did not differ, while up to dp = 150 mmHg, the valve with the leaflet of the original geometry (cycle 1) had 1.8 times lower BF values than in the case of the deformed and straightened leaflet (cycles 2 and 3).

To increase the stiffness of the leaflet, its titanium frame was replaced by the steel (abbreviation: ZNP-S). This operation significantly improved the valve properties: under the pressure of dp = 300 mmHg, the leaflet deformed to a lesser extent than the ZNP-Ti valve leaflet and the deformation itself was not permanent (Figure 14a,b). In subsequent work cycles, the ZNP-S valve was characterized by about 50% lower BF values than the ZNP-Ti valve (Figure 14, on the right side).

The study of the outflow valve with a titanium frame (abbreviation: ZWP-Ti) showed that in the first cycle of operation (just after manufacturing) already at a pressure of dp = 45 mmHg, the valve leaflets break and open backwards in the direction of flow (Figure 15a,b), which is a damage to the valve. In the second cycle of work, the fracture of the valve leaflets occurred at a pressure of 30 mmHg (Figure 15, on the right side). The same effect, but for a pressure of dp = 120 mmHg occurred for the valve with a steel frame (abbreviation: ZWP-S).

To solve this problem thickness of petal was increased and additional mechanical support surfaces were added into the valve ring. For such a modified valve with a titanium (abbreviation: ZWP2-Ti) and steel (abbreviation: ZWP2-S) stiffening frame, the leaflet fracture pressure was 320–340 mmHg, i.e., 2.8 times higher than for the ZWP-S valve and 7.5 times higher than for the ZWP-Ti valve. At the same time, the ZWP2-Ti valve showed a tendency to spontaneously restore the correct position of the leaflets after a slight flow towards normal operation. The reversal of the ZWP2-S valve leaflets was intrinsically irreversible.

As mentioned in Section 3.1, special technological retrograde flow channels were made in the ring of the outflow valve, through which blood flows backwards during the valve closure phase in order to better wash the valve leaflets and prevent the formation of blood clots. The influence of technological retrograde flow channels on the BF values of the ZWP-S valves is shown in Figure 16. The valve with completely open channels practically did not fulfill its hydrodynamic function (BF = 5 L/min for dp = 100 mmHg), while the closing of the channels reduces the BF 2.5 times (Figure 16, ZWP2-S curves). This observation was the basis for introducing another design change (ZWP3-S), consisting in the addition of flexible, polyurethane flow directing elements to the channels, operating in such a way that they close during reverse flow and open when the flow opens the valve. The characteristics of this kind of valve (Figure 16, ZWP3-S curve) show that the value of BF = 0.8 L/min is practically constant up to the pressure of 240 mmHg and then it increased: first abruptly and then monotonically. This effect is caused by the breaking of additional flow directing elements.

### 3.4. CLSM Analysis of the Blood Material Interaction after the Artificial Patient Model

The fluorescently labeled monoclonal antibodies anti-CD62P (to detect active platelets) and anti-CD45 (to detect active leucocytes) were used to characterize the blood clotting activation cascade on the surfaces of the valves. Fibrinogen is a hepatic plasma protein that is involved in the final coagulation procedure and is converted into a fibrillar protein—fibrin that coforms a blood clot. Fibrinogen combines with GpIIb/IIIa receptors to aggregate activated thrombocytes. It is classified as acute phase proteins. The Von Willebrand factor is an essential blood component involved in its coagulation process, a glycoprotein composed of several subunits encoded by a gene located on the chromosome 12. The von Willebrand factor is necessary for the adhesion of collagen platelets to the tested surfaces.

The surface coverage by activated platelets and leucocytes was determined (Figure 17).

Activation of the CD62P and CD45 antibody was much higher for the outflow than the inflow valve. In the case of fibrinogen and von Willebrand factor, the relationship was reversed. Greater activation for the inflow valve, smaller for the outflow valve. The effect observed may indicate an increased activation of blood on the inflow valve and an increased likelihood of clotting on the outflow valve.

### 3.5. In-Vitro Hemolysis and Thrombogenicity Assessment of ReligaHeart EXT Blood Pump with Composite Petal Valves

The blood flow through the inflow, single-leaflet valve and the outflow, bileaflet valve with additional flaps in the technological flow channels are shown in Figure 18a,b, respectively. The examinations were carried out on ovine blood. The valves were mounted in the ReligaHeart EXT ventricular assist device, so it was not possible to measure the pressure drop across the valves in this experimental setup. The pressure curve shown in the figures is the pneumatic pressure that drives the blood pump.

The effect of the valves on the erythrocytes (hemolysis) was assessed according to the methodology described in [23] determining the normalized index of hemolysis (NIH). The results are summarized in Table 1. The effect of the tested valves on erythrocytes was the same as that of disc valves (NIH = 0.019 g/100 L) and comparable with the hemolysis of the widely used clinically rotary blood pumps.

Valve thrombogenicity was assessed in *n* = 5 in-vitro examinations performed on sheep blood sampled intravenously with heparin-blocked coagulation. The blood was purchased in the Centre for Experimental Medicine, which provides a service for the sale of ram’s blood collected intravitally from donors (animals) specially farmed for this purpose, which does not require the consent of the Ethics Committee. The valves were part of the ReligaHeart EXT blood pump. During blood circulation, binding of heparin to antithrombin was gradually and spontaneously broken down, which restored the activity of the coagulation system. It has been proven [24] that as a result of this procedure, thrombus with a location and morphology similar to that formed in these pumps after long-term use on a living object are formed in the tested blood pump. The parameter measured to assess the activity of the coagulation system was the activated clotting time (ACT).

After finishing of the experiment, the blood pumps were dissected and the thrombotic material was assessed. All ZWP3-S and ZNP-S valves were thrombus-free (Figure 19a). In one case (the ZWP-Ti valve), a small clot stuck at the site of the defect of the leaflet surface was observed (Figure 19b). The effect of fibrin adhesion to damaged surfaces is a typical phenomenon and can be eliminated by improving the technology of manufacturing the valves themselves and their assembly in the target blood pump.

The conducted physical and biological tests showed that the inflow ZNP-S version of the composite valve and the outflow of the ZWP3-S version of the valve had little effect on the blood, both with regard to the coagulation system and erythrocytes.

## 4. Discussion

Flexible valves in heart assist pumps are known. However, the well-known disadvantage of transferring a valve similar to the anatomy of the human aortic valve is the high risk of thrombus formation in the bottom of the petals, caused by valve architecture that does not reflect biological conditions. The single-leaf artificial valve placed in the path of blood flow to the pulsating heart prosthesis has minimal effect on the flowing stream of fluid. This type of valve allows one to maintain laminar or transient blood flow and to avoid flow turbulence. This in turn reduces the shear stress on the blood cells, so that neither hemolysis nor excessive activation of platelets occurs. Platelet activation leads to the formation of blood clots, especially during long-term (more than 3 months) cardiac support. For these reasons, a single-leaflet valve, especially with a deformable polyurethane leaflet, has advantages over mechanical, tilting disc valves.

The conducted biocompatibility and functionality tests of the prototypes of pulsating VAD equipped with the newly developed, composite, petal valves confirmed high biocompatibility of these devices. However, they also pointed to extreme precision needed in their production and a proper setting in the blood pumps. Therefore, it should be stressed that the developed solution has demonstrated all the required functional properties, including its full biocompatibility. However, further work is necessary in order to optimize the manufacturing technology and assembly procedures of the valves in the scope of even small scale market production.

Performed tests of the valve operation indicated that it needs a stronger mechanical support for the leaflets in the direction of reverse flow. This remark applies in particular to the outflow valve, which, due to the pressure load on the pump, is exposed to large pressure gradients. The construction of the inflow valve has been modified too with the aim to increase the support surface necessary for the leaflet and protect it from incidental forcing it into the lumen of the valve channel under the pressure acting on it during the closing phase. The initial leaflet structure had a variable support surface extending along both lateral free edges and disappearing towards the bottom of the leaflet. Some solution to this problem may be adding to the leaflet structure the support surface in the form of polyurethane layer. The thickness of this layer may be differentiated (gradient, growing towards the bottom of the flake), which will allow one to increase its strength and stabilize the flake during the closing phase and closed the valve.

The modified structures of both inflow and outflow valves were implemented in extracorporeal, pulsating blood pump ReligaHeart EXT. The pump prepared in this way was subjected to fundamental tests in contact with blood. Hemolysis studies have confirmed that the structure of the blood flow through the valves (especially the outflow valve with additional channels not found in other types of valves) does not damage erythrocytes. The NIH values of the tested pump are comparable with those used clinically.

In terms of thrombogenicity, performed studies have shown that the main issue influencing the formation of thrombus is the quality of the surface contacting with blood. In the tested pumps, thrombi formed in places where the leaflets were stuck to the rings, which made the surface in this place not smooth enough. Nevertheless, considering that the tested specimens were in fact hand-made prototypes, the atrombogenic properties of the valves should be considered good.

Two flushing channels located at the bottom of ring of the inflow valve are equipped with additional miniature petals closing the flow through the channels in the valve closing phase. They allow the bottom of the main leaflets to be flushed in the closing phase of the valve. As a result, the main function of the leaflets (especially the bottoms, which form the blood stasis pocket after closing the valve) has been implemented, which is essential for preventing thrombus formation in the leaflet bottom without reducing the functionality and flow and valve tightness parameters.

## 5. Conclusions

In conclusion, the results of the work confirmed the innovative properties of ReligaHeart EXT pulsatile heart assist pump construction equipped with elastic, composite valves: single-leaflet at the inflow and double-leaflet with flushing channels of the outflow. The construction of petals and valve rings was designed in such a way that the valve leaflets form integrated surfaces with surrounding parts of the chamber, ensuring the washing effect of the valve petals with blood on their smooth surfaces.

The developed concept for the valves provides:Integration of the surface of the leaflets with the adjacent surfaces of the heart prosthesis,Smooth movement of the leaflets, allowing positioning during the optimal opening, which forms a laminar blood stream,Controlled elasticity of the leaflets, which ensures their correct mechanical work in the prosthesis: effective opening and tight soft closing—without unnecessary shear or crushing stress for the blood cells.

## Figures and Tables

**Figure 1 polymers-12-02986-f001:**
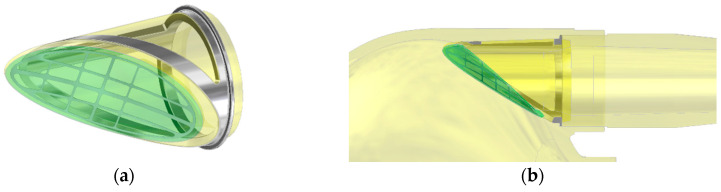
Inflow valve (**a**) and cross section, (**b**) through the flow channel area.

**Figure 2 polymers-12-02986-f002:**
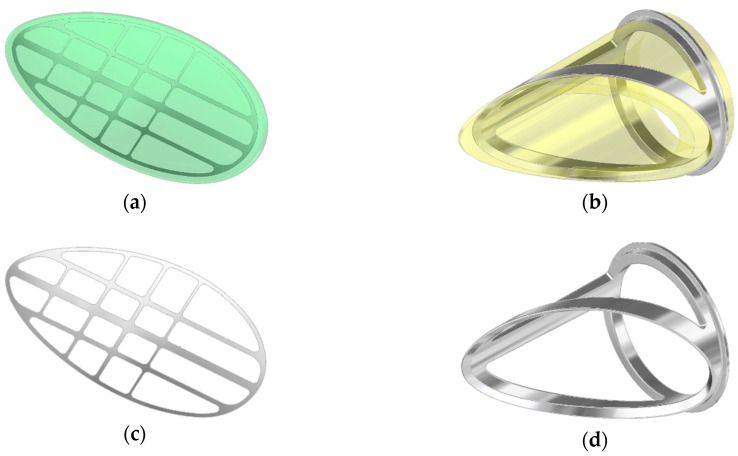
Inflow valve construction details: (**a**) composite leaflet, (**b**) polyurethane valve ring, (**c**) open-shape frame, (**d**) the metal frame.

**Figure 3 polymers-12-02986-f003:**
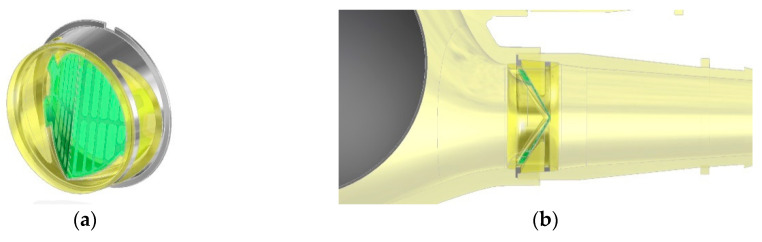
Outflow valve (**a**) and cross section (**b**) through the flow channel area.

**Figure 4 polymers-12-02986-f004:**
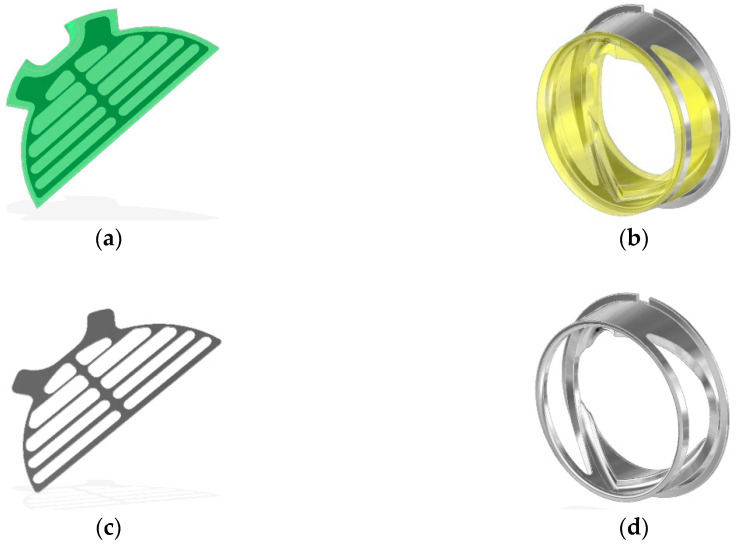
Outflow valve construction details: (**a**) composite leaflet, (**b**) polyurethane valve ring, (**c**) open-shape frame, (**d**) the metal frame.

**Figure 5 polymers-12-02986-f005:**
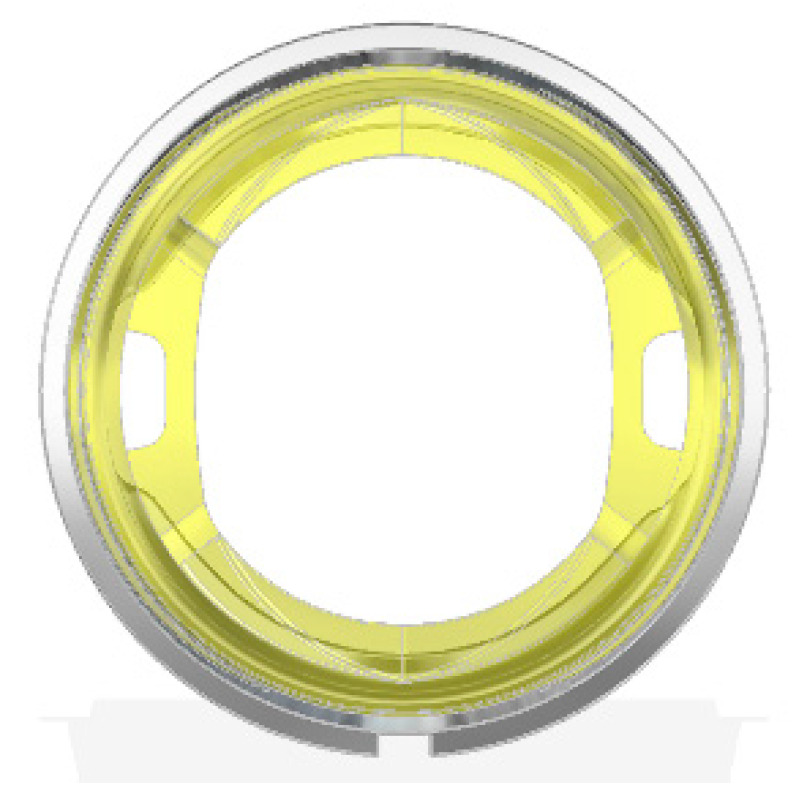
Outflow valve ring with channels improving the leaflets washing.

**Figure 6 polymers-12-02986-f006:**
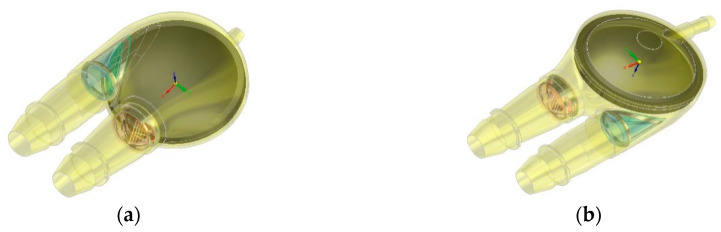
ReligaHeart EXT heart assist pump with new valves (**a**) view from the side of the blood chamber, (**b**) view from the pneumatic chamber.

**Figure 7 polymers-12-02986-f007:**
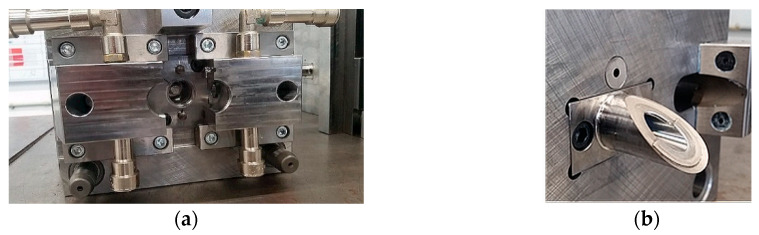
Injection molds for the ring of the inflow valve: (**a**) mold seat and (**b**) mold stamp.

**Figure 8 polymers-12-02986-f008:**
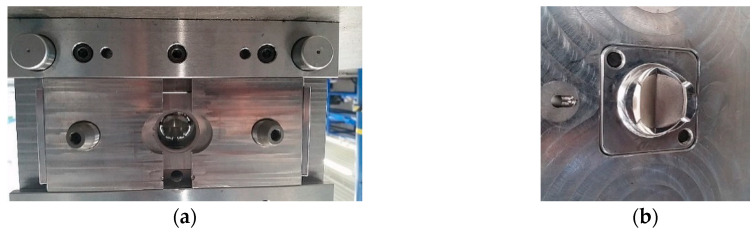
Injection mold for the ring of the outflow valve: (**a**) mold seat and (**b**) mold stamp.

**Figure 9 polymers-12-02986-f009:**
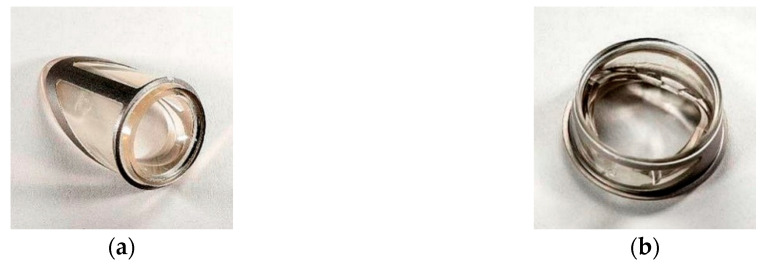
Polymer components of the valve rings: (**a**) the inflow and (**b**) outflow valve.

**Figure 10 polymers-12-02986-f010:**
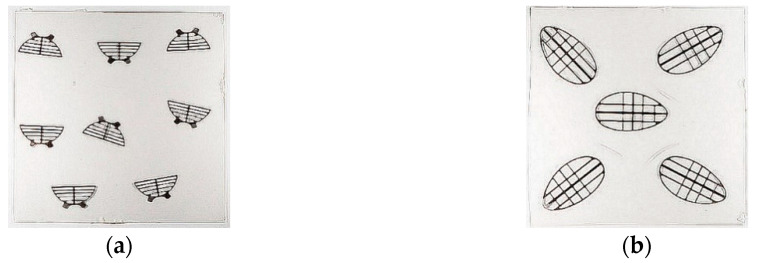
Foils with stiffening frames made of titanium after pressure processing for (**a**) outflow and (**b**) inflow valves, respectively.

**Figure 11 polymers-12-02986-f011:**
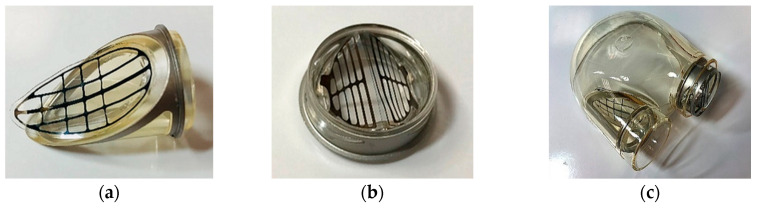
Assembled (**a**) inflow and (**b**) outflow valves, (**c**) blood chamber of the ReligaHeart EXT ventricular assist device (VAD).

**Figure 12 polymers-12-02986-f012:**
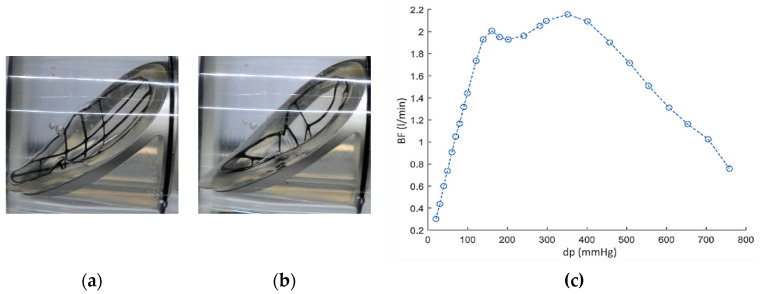
ZNP-Ti valve: leaflet deformation for (**a**) dp = 100 mmHg, (**b**) dp = 760 mmHg and (**c**) influence of leaflet deformation on valve sealing (visible for dp > 400 mmHg).

**Figure 13 polymers-12-02986-f013:**
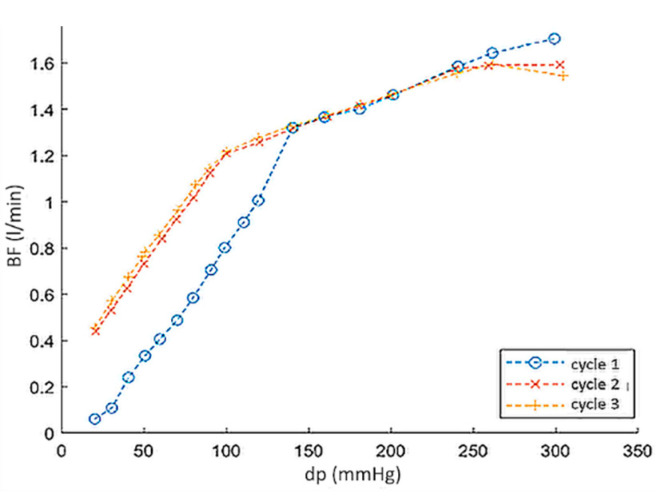
Outflow valve ring with channels improving the leaflets washing.

**Figure 14 polymers-12-02986-f014:**
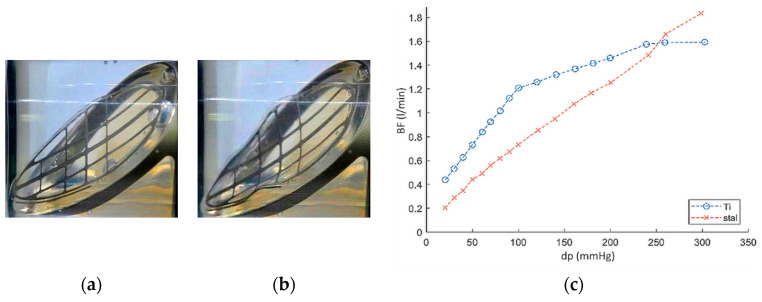
ZNP-S valve: leaflet deformation for (**a**) dp = 10 mmHg, (**b**) dp = 300 mmHg and (**c**) comparison of backflow (BF) for inflow valves with the frame made of titanium (Ti) and steel (S).

**Figure 15 polymers-12-02986-f015:**
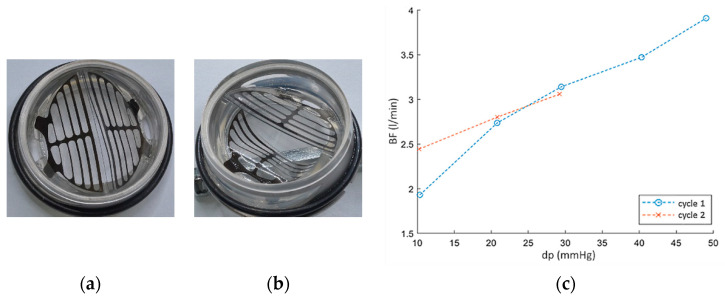
ZWP-Ti valve: (**a**) leaflet after manufacturing, (**b**) leaflets break and open backwards in the direction of flow for dp = 45 mmHg, (**c**)the pressure at which the leaflets of the valve break for first and second cycle of work.

**Figure 16 polymers-12-02986-f016:**
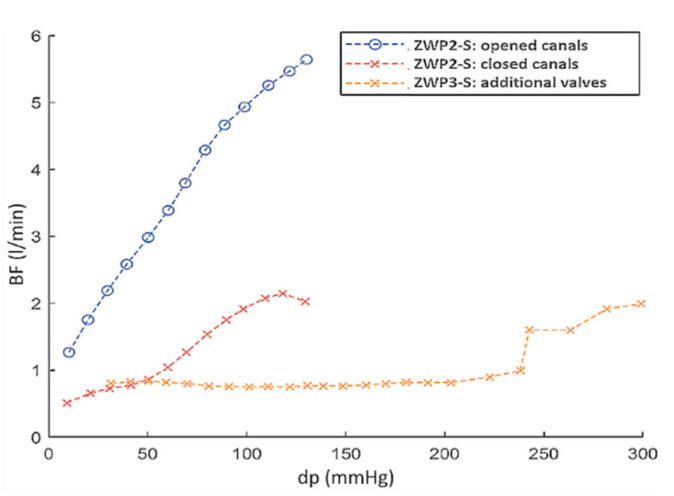
Comparison of backflows of various types of outflow valves (description in the text).

**Figure 17 polymers-12-02986-f017:**
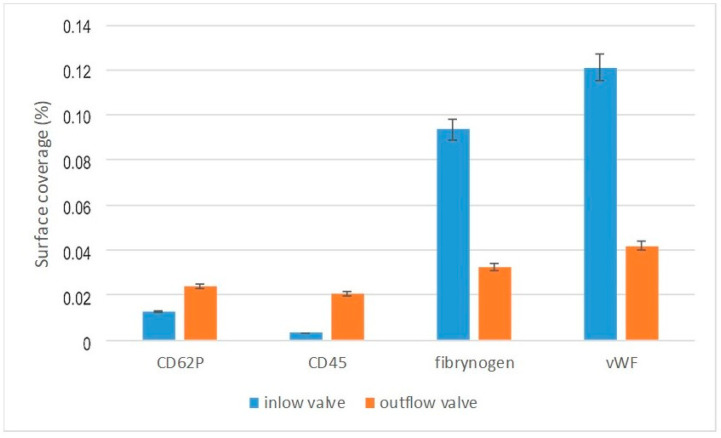
Blood–material interaction after the “artificial patient” test.

**Figure 18 polymers-12-02986-f018:**
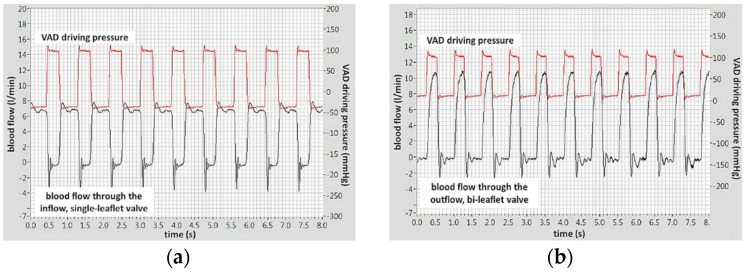
Blood flow and VAD driving pressures waveforms: (**a**) through the inflow valve, (**b**) through the outflow valve. Hemodynamic conditions: VAD afterload pressure = 100 mmHg and heart rate = 50 BPM.

**Figure 19 polymers-12-02986-f019:**
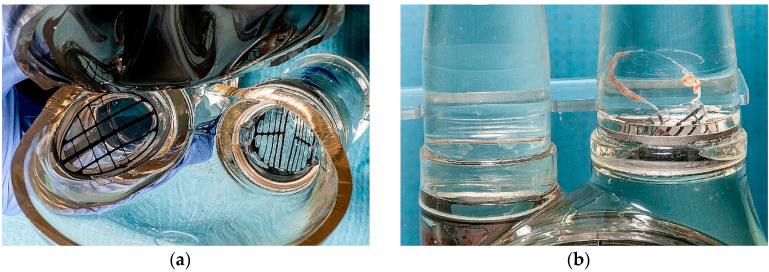
View of valves after in-vitro thrombogenicity study: (**a**) view from the inside of the blood chamber on the inflow and outflow valves, (**b**) side view at the place where the valves are glued in.

**Table 1 polymers-12-02986-t001:** Comparison of hemolysis of pulse pumps with the tested valves and hemolysis of reference pumps.

Type of Blood	Type of Valve	NIH (g/100 L)
human	ZWP-Ti: wrong operation of outflow valve due to leaflet fracture	0.103
ovine	ZWP3-S: proper operation of valves (*n* = 2)	0.019
human	reference: ReligaHeart EXT extracorporeal, pulsating with tilting disc valves (*n* = 6)	0.019
human	reference: HVAD implantable rotary blood pump (*n* = 2)	0.017
human	reference: BP80 extracorporeal rotary blood pump (*n* = 7)	0.012

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
