# Peer review of "Design, Manufacturing Technology and In-Vitro Evaluation of Original, Polyurethane, Petal Valves for Application in Pulsating Ventricular Assist Devices"

_polymers, 2020, doi:10.3390/polym12122986_

Round 1

Reviewer 1 Report

Major et al presented a study that aimed at minimizing life-threatening thromboembolic adverse effects and blood damage in VADs intended for pediatric patients through applying a new biomimetic heart valve design based on metal-polymer composites and flexible leaflets.

- what is the advantage of the single leaflet valve? and what's the reasoning behind it?

- What manufacturing modality was used to manufacture it?

- What were the boundary conditions used for the simulation?

- Was the simulation done with the valves? was FSI implemented?

- The figures legends do not show. Please provide larger figure legends.

- The authors are encouraged to convert their pressure findings from bar to mmHg to relate to units adopted in clinic.

- in the hydrodynamic testing, how did the curves look versus time? both the flow and the pressure gradient?

Author Response

All remaks given by the Reviewer has been considered. The detailed answers has been attached

Reviewer 2 Report

The article “polymer-1002513” entitled “ Design, manufacturing technology and in-vitro evaluation of original, polyurethane, petal valves for application in pulsating ventricular assist devices” was interesting and structural integrity. The authors used specially engineered moulding thermoplastic polyurethane inserts onto a rubber backing layer to increase service life. By the way, using numerical simulation method to simulating valve function and “artificial partient” to determine hydrodynamic properties of the valve in the environment similar to physiological conditions. However, do the authors conduct a cyclic loading test, or what is the life span of this device?

Special

  1. The resolution of Figure .1 and Figure 2 was not enough.
  2. The conclusion format should be fully justified.
  3. Please spell out the numerical simulation method, software, and
    parameter setting.
  4. The scale bar in figure 7~11 was not clear.
  5. From line 225 to line 251, the author mentioned the numerical simulation of this device in the high-pressure injection process, why did authors describe the content of this simulation work? Please illustrate the relevant discussion or literature comparison in this article.

Author Response

(The authors gave the same response as above.)
